# Mechanical Properties and Corrosion Behavior of Ti6Al4V Particles Obtained by Implantoplasty: An In Vitro Study. Part II

**DOI:** 10.3390/ma14216519

**Published:** 2021-10-29

**Authors:** Jorge Toledano-Serrabona, Maria Ángeles Sánchez-Garcés, Cosme Gay-Escoda, Eduard Valmaseda-Castellón, Octavi Camps-Font, Pablo Verdeguer, Meritxell Molmeneu, Francisco Javier Gil

**Affiliations:** 1Bellvitge Biomedical Research Institute (IDIBELL), Department of Oral Surgery and Implantology, Faculty of Medicine and Health Sciences, University of Barcelona, 08907 Barcelona, Spain; jorgetoledano25@gmail.com (J.T.-S.); cgay@ub.edu (C.G.-E.); eduardvalmaseda@ub.edu (E.V.-C.); ocamps@ub.edu (O.C.-F.); 2Bioengineering Institute of Technology, International University of Catalonia, 08195 Barcelona, Spain; pverdeguerm@gmail.com; 3Biomaterials, Biomechanics and Tissue Engineering Group (BBT), Department of Materials Science and Engineering, Polytechnic University of Catalonia, 08019 Barcelona, Spain; meritxell.molmeneu@upc.edu; 4Faculty of Dentistry, International University of Catalonia, 08195 Barcelona, Spain

**Keywords:** implantoplasty, corrosion, Ti6Al4V, dental implant

## Abstract

In the field of implant dentistry there are several mechanisms by which metal particles can be released into the peri-implant tissues, such as implant insertion, corrosion, wear, or surface decontamination techniques. The aim of this study was to evaluate the corrosion behavior of Ti6Al4V particles released during implantoplasty of dental implants treated due to periimplantitis. A standardized protocol was used to obtain metal particles produced during polishing the surface of Ti6Al4V dental implants. Physicochemical and biological characterization of the particles were described in Part I, while the mechanical properties and corrosion behavior have been studied in this study. Mechanical properties were determined by means of nanoindentation and X-ray diffraction. Corrosion resistance was evaluated by electrochemical testing in an artificial saliva medium. Corrosion parameters such as critical current density (icr), corrosion potential (E_CORR_), and passive current density (i_CORR_) have been determined. The samples for electrochemical behavior were discs of Ti6Al4V as-received and discs with the same mechanical properties and internal stresses than the particles from implantoplasty. The discs were cold-worked at 12.5% in order to achieve the same properties (hardness, strength, plastic strain, and residual stresses). The implantoplasty particles showed a higher hardness, strength, elastic modulus, and lower strain to fracture and a compressive residual stress. Resistance to corrosion of the implantoplasty particles decreased, and surface pitting was observed. This fact is due to the increase of the residual stress on the surfaces which favor the electrochemical reactions. The values of corrosion potential can be achieved in normal conditions and produce corroded debris which could be cytotoxic and cause tattooing in the soft tissues.

## 1. Introduction

Commercially pure titanium (cp-Ti) dental implants are an excellent long-term treatment for patients with loss of teeth [1,2]. Titanium dental implants (grades I–IV) are highly reliable due to their excellent biocompatibility, mechanical characteristics, and good corrosion resistance among other properties [3,4,5]. However, the strongest grade of cp-Ti has a strength of around 550 MPa. Thus, other Ti alloys have been designed in order to increase the strength of the material [6]. Among Ti alloys, Ti6Al4V is an (α-β)-type that has been used in a wide range of biomedical purposes. This alloy was first developed in the aerospace industry but due to its strength, excellent corrosion resistance, and biocompatibility, Ti6Al4V is used for biomedical purposes. There are some concerns related to the long-term use of Ti6Al4V that have already been outlined in Part I. In summary, as Ti6Al4V contains toxic V, different alloys such as Ti6Al7Nb, Ti5Al2.5Fe, or TiZr have been proposed to replace it. However, Ti6Al4V remains the most widely used Ti alloy for dental implants [6,7,8,9,10,11]. 

During the lifespan of dental implants, different complications may occur, such as mechanical, biological, and esthetic complications. Biological complications comprise peri-implant mucositis and peri-implantitis, both of which are inflammatory conditions induced by bacterial plaque affecting the surrounding tissues of the dental implant [12]. Nonsurgical treatment is sufficient for remission of peri-implant mucositis, but not for peri-implantitis. Depending on the type of peri-implant defect and the location of the implant, surgical treatment of peri-implantitis includes access surgery, resective surgery, peri-implant bone reconstruction (regenerative or reconstructive surgery), or a combination of these techniques [13]. Additionally, it is crucial to decontaminate the dental implant surface during nonsurgical or surgical procedures to stop the progression of the disease [14].

Metal particles of different sizes are generated during the insertion of the dental implant, bed preparation, machining to improve the fit of the prothesis, or wear due to micro-movements or functional loading. Dental implant surface decontamination procedures to treat peri-implantitis can also generate metallic debris. These particles with high internal energy can have significant physiological effects, such as an increase of the corrosion rate with generation of debris, cytotoxicity, an increase of ion release, and loss of the mechanical properties (such as crack nucleation on the surface resulting in fatigue). These aspects might disrupt osseointegration and cause bone resorption (osteolysis), which in turn may lead to implant loss [15,16,17,18,19]. 

The oral medium, with the presence of saliva, bacteria, other metals and alloys, and chemical products (for instance, gastric acids caused by reflux) causes corrosion and chemical degradation of titanium or titanium alloys [20,21,22,23,24]. 

In addition to the negative impact on biology, the corrosion of dental alloys can also have a negative effect on function and aesthetics of a dental prosthesis. The process of corrosion generates corrosion debris, which contains toxic oxides and metal ions that may not only come into contact with the surrounding cells and tissues, but also be distributed throughout the body through the bloodstream, intestines, and urinary excretory system. Debris particles of 10 to 20 μm in size have been detected at the implant surface and peri-implant bone, and distant sites, such as the lungs, liver, and kidney [25].

The purpose of this investigation was to describe the mechanical characteristics, including hardness and elastic modulus, as well as to determine the corrosion behavior of metal particles originating from commonly used Ti6Al4V (grade 5) following an implantoplasty procedure. This aims to raise the awareness of potential detrimental effects of implantoplasty and the need for careful consideration of the dental implant material.

## 2. Materials and Methods

### 2.1. Sample Preparation

A single investigator (J.T-S.) carried out implantoplasty procedures of Ti6Al4V dental implants following the drilling protocol described in previous publications [26,27]. A GENTLEsilence LUX 8000B turbine was used (KaVo Dental GmbH, Biberach, Germany) with water irrigation at room temperature. The surface was sequentially modified with a fine-grained tungsten carbide bur and two polishers, as described in Part I. The sample was lyophilized to rid the water from the metal particles.

### 2.2. Scanning Electron Microscopy and Mechanical Properties

As previously described in Part I, the morphometry of the sample was determined by scanning electron microscopy Neon 40 Surface Scanning Electron Focused Ion Beam Zeiss (Zeiss, Oberkochen, Germany).

A hardness analysis was performed using nanoindentation techniques on the dental implant (base material) and on the metal debris released during implantoplasty in order to determine the hardness and elastic modulus of both types of samples.

The nanoindentation assays were carried out using “Berkovich” type indenters, with a constant strain rate of 0.05 s^−1^. An iMiro (KLA tencor, Kavo dental, Bibereach, Germany) and a Nanoindenter XP (MTS Systems Corporation, Oak Ridge, TN, USA) were used for the determination of implant hardness and metal debris hardness, respectively.

Residual stresses were measured with a diffractometer incorporating a Bragg–Brentano configuration (D500, Siemens, Wurzburg, Germany). Following Bragg’s law, the superficial stress can be calculated since X-Ray diffraction allows the determination of the interplanar distance before and after shot blasting. After the treatment, the interplanar distance is smaller due to the residual compressive stress. The differences in the interatomic distance allow calculation of the microstrain, which, together with the elastic modulus of the material, allows the determination of the residual stress on the surface. The measurements were done for the family of planes (213) which diffracts at 2θ = 139.5°. The elastic constants of Ti at the direction of this family of planes are EC = (E/1 + υ) (213) = 90.3 GPa [1,4]. Eleven ψ angles, 0°, and five positive and five negative angles were evaluated. The position of the peaks was adjusted with a pseudo-Voigt function using appropriate software (WinplotR, free access online), and then converted to interplanar distances (dψ) using Bragg’s equation. The dψ vs. sen2ψ graphs and the calculation of the slope of the linear regression (A) were done with appropriate software (Origin, Microcal, Northampton, MA, USA). The residual stress is: σ = EC × (1/d_0_) × A; where d_0_ is the interplanar distance for ψ = 0° [28,29]. 

### 2.3. Corrosion Test

We prepared ten discs of as-received material (Ti6Al4V) used for machine dental implants (control group), and ten discs of Ti6Al4V cold-worked at 12.75% in order to achieve the same mechanical properties (hardness, strength, Young’s modulus, strain, and residual stress) as the implantoplasty particles (experimental group). This preparation was aimed at increasing the reliability of corrosion tests. The tests with the particles included in polymeric non-conductive resin showed high deviations due to the lack of continuity of the metallic particles. 

All discs were polished metallographically following the recommendations defined in ASTM E3-17 Standard [30]. Discs were treated with sequential grinding steps with different SiC papers. Samples were finally polished using diamond suspension paste with an average particle grain size ranging from 5 μm to 0.1 μm (Buehler S4, Lake Bluff, IL, USA). All metallic disc-shaped samples were smoothed up to a surface roughness (Ra) under 20 nm. Upon completion of the polishing phase, samples were cleaned with a sequential immersion bath protocol using cyclohexane, isopropanol, ethanol, deionized water, acetone, and ethanol for 15 min for each cleaning bath, together with sonication (all chemicals from Sigma Aldrich, St. Louis, MO, USA). 

Testing sample groups were kept individually immersed in a constant volume of electrolyte for all the measurements. Hank’s solution was selected as an electrolyte in order to simulate the real oral physiological conditions (composition displayed in Table 1). The electrolyte was kept under constant pH (6.7) during the experiments and was completely renewed for each experiment [31,32,33]. 

The tests were carried out with a PARSTAT 2273 potentiostat (Princeton, San Jose, CA, USA) controlled by Voltamaster 4 software (Radiometer Analytical, Villeurbanne Cedex, France). For both open circuit potential measurement tests and potentiodynamic tests, the reference electrode was a calomel electrode (saturated KCl), with a potential of 0.241 V versus the standard hydrogen electrode. The auxiliary electrode was a platinum electrode with a surface of 240 mm^2^ (Radiometer Analytical, Villeurbanne, France). All the tests were carried out in a Faraday box to avoid the interaction of external electric fields. Figure 1 depicts the experimental setup of corrosion test.

#### 2.3.1. Open Circuit Potential 

Tests were carried out for 5 h for all samples, taking measurements every 10 s. The potential was considered to be stabilized when the variation of the potential was less than 2 mV for a period of 30 min as indicated by the ASTM G5 and ISO 10993-5:2009 standards [31,32]. This test determines the susceptibility to corrosion (lower potential). The data and E–t curves were obtained using the PowerSuite program with the PowerCorr-Open circuit function. 

#### 2.3.2. Potentiodynamic Tests (E-log^(I)^ Curves)

Cyclic potentiodynamic polarization curves were obtained for the two study groups following the ASTM G5 standard. In this test, a potentiostat induced a variable electrical potential between the sample and the reference electrode, thus causing the passage of a current between the sample and a platinum counter electrode. 

Before starting the test, the system was allowed to stabilize by means of a 5 h open-circuit test. After stabilization, a potentiodynamic test was made by means of a cyclic potential range from −0.8 mV to 1.7 mV at a rate of 2 mV/s. These parameters were analyzed by the PowerSuite program, using the PowerCorr-Cyclic Polarization function to obtain the curves. 

The parameters obtained were corrosion current density (*i_CORR_* (µA/cm^2^)) and corrosion potential (E_CORR_ (mV)) (value at which the current density changes from cathodic to anodic). The E_CORR_ and I_CORR_ parameters were obtained by extrapolation of the Tafel slopes [34,35,36,37,38,39].

According to ASTM G102-89, obtaining these values allows calculation of the polarization resistance (*R_p_*) by means of the Stern–Geary expression (Equation (1)) and the corrosion rate (*C_R_* in mm/year) (Equation (2)).
(1)$$R_{P}=\frac{\beta_{a}\beta_{c}}{2.303(\beta_{a}+\beta_{c})t_{corr}}$$
(2)$$CR=K_{1}\frac{t_{corr}}{\rho }EW$$
where *K*_1_ = 3.27 × 10^−3^ mm-g/μA-cm-year, the density of Titanium is 4.54 g/cm^3^, *EW* is its equivalent weight (11.98, it is considered dimensionless in these calculations), β*_a_* is the slope of the anodic curve, and β*_c_* the cathodic one. The polarization resistance indicates the resistance of the sample to corrosion when subjected to small variations in potential.

### 2.4. Statistical Analysis

Statistically significant differences were studied using statistical software (MinitabTM 13.1, Minitab Inc., State College, PA, USA). ANOVA tables with multiple comparison Fisher tests were calculated. The level of significance was established at *p*-value < 0.05.

## 3. Results

### 3.1. Scanning Electron Microscopy and Mechanical Properties

As described in Part I, the metal particles obtained by implantoplasty had a flattened geometry similar to “flakes” with clear signs of high levels of plastic deformation resulting from chipping (Figure 2a). The microstructure observed by SEM of the particles showed slip bands revealing the Widmanstatten plate microstructures of the Ti6Al4V alloy, which indicates that the Ti6Al4V absorbed a large deformation before the fracture. Figure 2b shows the Widmanstatten microstructure with very thin lamellar structure. The plates consisted of alpha phase plates (white phase) surrounded by beta phase plates (dark phase) [40,41,42].

Arrays of 100 indentations (10 × 10) at different loads (1, 2, 3, and 4 mN) were carried out. The hardness and elastic modulus were constant for the entire range of applied loads, as shown in Table 2. The subsequent tests (hardness and modulus of elasticity maps) were carried out at a constant load of 2 mN. At this load, the tip defect (rounding due to the manufacturing process) and the effect of polishing-induced roughness are negligible. Thus, the minimum load that allowed a correlation of mechanical properties and microstructure was selected. No statistically significant differences are obtained in the hardness and elastic modulus.

Figure 3 depicts the hardness and elastic modulus distribution maps measured on the inner and threaded surface areas of the dental implant. The comparative analysis of the results showed a joint increase in the hardness and elastic modulus of the base material (Ti6Al4V) in the threaded area as a consequence of the plastic deformation generated during the machine manufacturing processes.

The hardness and elastic modulus curves as a function of penetration depth determined for the metal debris is depicted in Figure 4. 

Implantoplasty increased the hardness and elastic modulus of the metallic particles. This fact is explained because milling increased the density of dislocations and defects in the metallic structure until the implant fractured in the form of particles. As expected, the estimation of the maximum deformation decreased, as did the toughness of the metal. Residual stresses were negative in all cases, which indicates a state of compression [43,44]. Thus, the implantoplasty procedure markedly increased residual stresses. 

Additionally, cold-working of 12.5% Ti6Al4V resulted in mechanical properties which are very similar to those of implantoplasty (Table 3). Thus, cold-worked disks simulate implantoplasty particles in corrosion studies. In all mechanical parameters the differences between control disks and either implantoplasty particles or cold-worked disks were significant (*p* < 0.05). 

### 3.2. Corrosion Behavior

Open corrosion potential (E_OCP_) is determined when a steady state is reached by the corrosion system, in which both cathodic and anodic reaction rates are properly balanced with no net current flow to or from the electrode. E_OCP_ value is used to qualitatively indicate the corrosion behavior of a material. It can be categorized as active or passive according to its sign [3,25]. 

The open circuit potential (E_OCP_) results are shown in Figure 5. The potential was on average −204 ± 18 mV for the titanium control, and −254 ± 3 mV (*p* < 0.05) for the implantoplasty samples, which means that the titanium control was more electropositive and in consequence more resistant to corrosion. 

The potentiodynamic curves of the two groups can be observed in Figure 6. For all the calculated parameters, the control group presents better corrosion resistance than the implantoplasty group, since its E_CORR_, I_CORR_, and corrosion rate values are lower, and the polarization resistance is higher (Table 4). However, these differences were not statistically significant for any of the mentioned parameters (0.15 > *p* > 0.05).

Figure 7 depicts SEM images of the surfaces of the control and implantoplasty samples after the corrosion tests. We selected areas with machining failures, as they are the most susceptible to corrosion. The implantoplasty sample showed much pitting, while the control sample did not display any [45]. 

## 4. Discussion

Nanoindentation tests showed that the Ti alloy particles released during implantoplasty were harder, probably because of the stress applied during machining. It is well known that when metals are machined, defects in the crystalline lattice (band slips, twins, etc.) increase the hardness of the metal to the point that it cannot absorb any more energy, causing it to fracture at those points. Thus, these particles have much higher internal energy than the rest of the dental implant. According to the laws of thermodynamics, the system will release energy to become more stable, thus being more susceptible to ion release and electrochemical corrosion; this can therefore generate non-cytocompatible oxides [3,46]. Although there is no joint wear in the oral cavity, implantoplasty milling procedures could have a similar effect, though not sustained over time. Moreover, in the oral cavity, the presence of physiological fluids and the stable temperature of 37 °C favor this release of ions and corrosion of the metal [28,47]. Therefore, from our point of view, the main risk of implantoplasty lies in the high internal energy of the metal debris produced, since its reaction with the medium to lower the internal energy levels may give rise to products or interactions that exert a toxic effect upon the physiological environment, especially human cells. Thus, implantoplasty might imply a risk of contributing to implant failure over the medium to long term, due to aseptic bone loss. In addition, there is also the risk of an increased presence of metal ions in the blood and their accumulation in the organs. 

As Figure 6 shows, there was an almost constant open corrosion potential over immersion time in all samples. This signal (potential) stability would be directly connected to the formation of a stable Ti-oxide passive film. The stability of this inert film depends to a great extent on the volume of the newly formed oxides. In the case of Ti6Al4V, there are several mixed oxides (non-stoichiometric oxides), which should be as similar as possible to the volume of the base-metal in order to protect the metal against corrosion. This happens particularly in the case of Ti6Al4V for all the oxides, which usually leads to the formation of thin passive films with oxide volumes very similar to the bulk metals, thus avoiding the formation of both cracks and breaks. This favors the passivity of the Ti6Al4V alloys. 

The corrosion potential results showed that control samples had the best corrosion resistance. As-received material presented a good chemical homogeneity as a result of an annealing treatment. In addition, annealing heat treatment avoided the presence of residual stresses, which could favor the decrease of galvanic corrosion rates. Implantoplasty samples showed a decrease in the corrosion potential due to the residual stress induced by the machining process on the surface [48,49,50,51,52], which can enhance surface chemical reactivity during corrosion testing. As is well-known, machining of metals induces residual stresses, which can affect their in-service behavior, as reported by several authors [53,54]. Corrosion resistance can be decreased by the presence of other metallic elements in the mouth (stainless steel wires for orthodontics, metals for prosthetics, among others) that create a galvanic current due to the presence of metallic materials of different chemical natures in an electrochemical environment. 

Finally, SEM shows the presence of pitting on the implantoplasty surfaces due to the electrochemical corrosion. Pitting involves a loss of material due to corrosion by migration of the reaction product (titanium oxide) into the physiological medium. These particles released as a result of corrosion and/or mechanical treatment, such as implantoplasty, have been reported to cause adverse allergic reactions in humans [55,56]. There is no current consensus on the risk of particles released from titanium implants; however, it would be prudent for clinicians to carefully evaluate the materials used, and to consider the potential risks of the individual constituents of any alloy, as indicated in this study.

## 5. Conclusions

Ti6Al4V alloy particles released during implantoplasty show higher hardness, mechanical strength, and compressive residual stresses than the control Ti6Al4V material. These compressive residual stresses, due to the higher deformation in the Widmanstatten microstructure, cause inferior corrosion behavior, both in open circuit potential and in potentiodynamic tests. Implantoplasty particles present worse corrosion resistances than the original samples, since their E_CORR_, I_CORR_, and corrosion rate values are higher, and the polarization resistance is lower. The increase in corrosion rate due to implantoplasty causes pitting on the surface of the samples. Clinicians must be aware of the potential risks of implantoplasty, because of a reduction in corrosion resistance among metal particles released during this procedure. 

## Figures and Tables

**Figure 1 materials-14-06519-f001:**
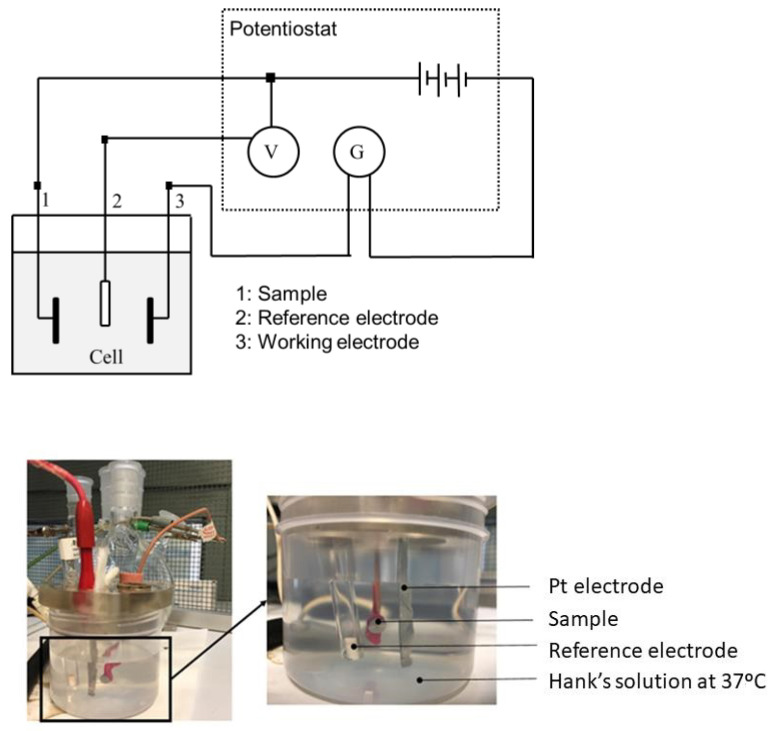
Three-electrode electrical circuit setup diagram and equipment used in electrochemical tests.

**Figure 2 materials-14-06519-f002:**
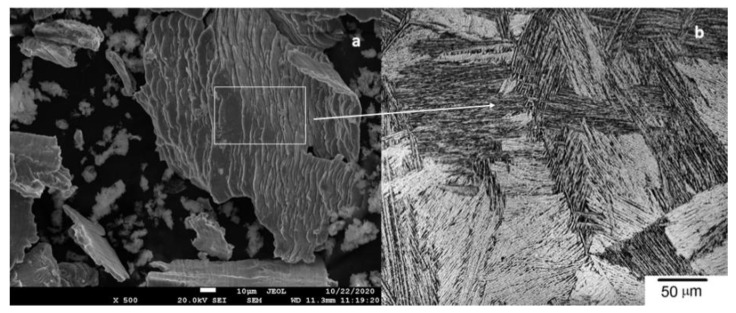
Scanning electron microscopy images: (**a**) implantoplasty particles at ×500 magnification; (**b**) microstructure of Widmanstatten of the particles.

**Figure 3 materials-14-06519-f003:**
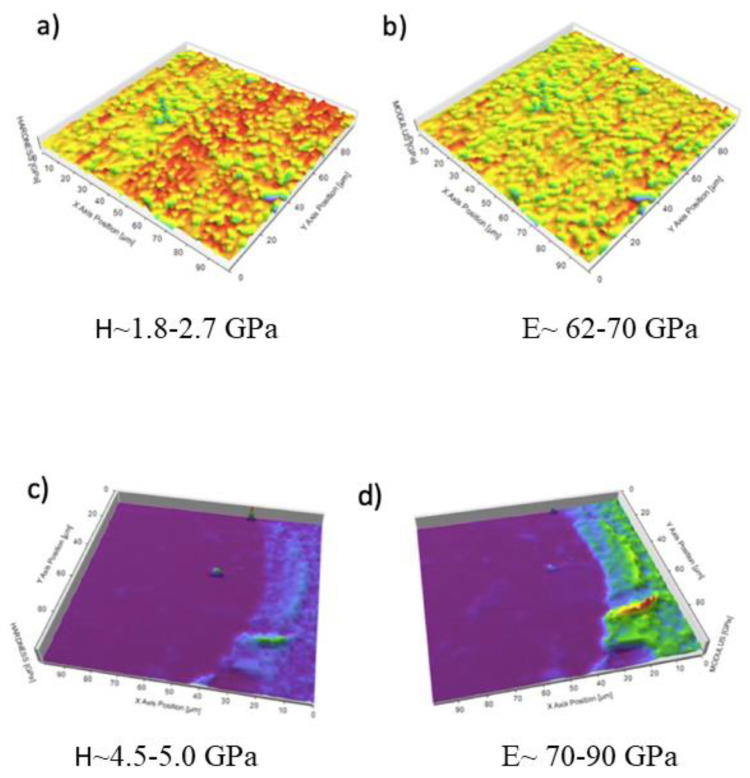
(**a**) Map of hardness distribution of the inner zone of the screw; (**b**) map of elastic modulus of the inner zone of the screw; (**c**) map of hardness of the surface zone of the screw; (**d**) map of elastic modulus of the surface of the screw. Figure 3 shows the hardness and elastic modulus curves as a function of penetration depth determined for the metal debris. Particles were harder than the base material. On the other hand, the elastic modulus of the metal debris showed values similar to those of the implant thread area.

**Figure 4 materials-14-06519-f004:**
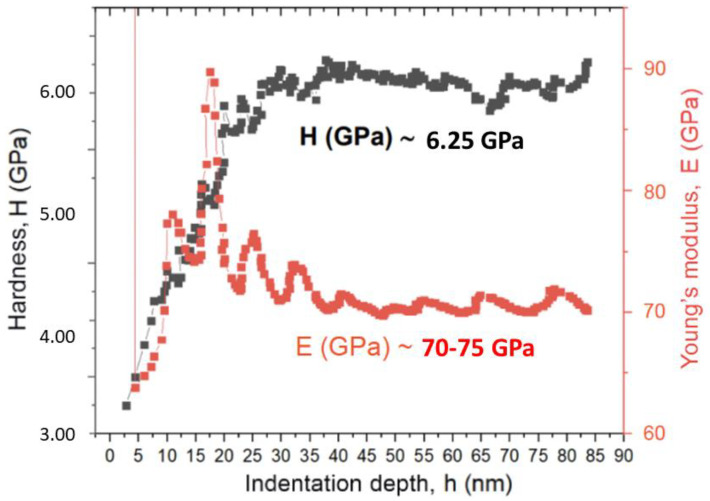
Hardness and elastic modulus curves according to indentation depth of the particles of the metal debris.

**Figure 5 materials-14-06519-f005:**
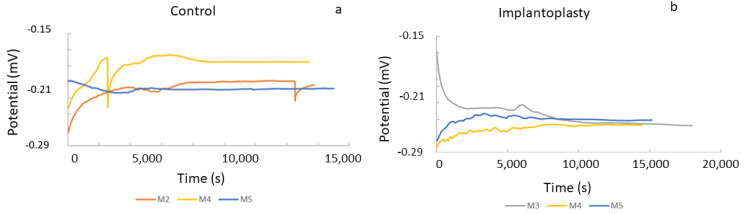
Curves of the open circuit potential for (**a**) control disc (original dental implant material) and (**b**) implantoplasty samples.

**Figure 6 materials-14-06519-f006:**
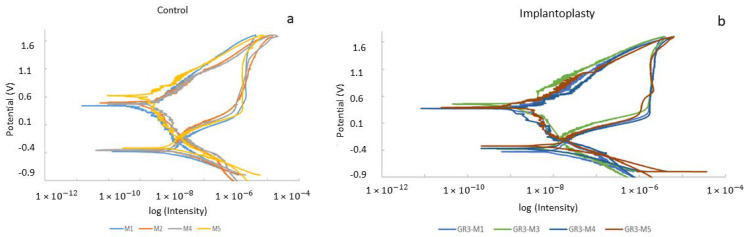
Potentiodynamic curves for (**a**) control disc (original dental implant material) and (**b**) implantoplasty samples.

**Figure 7 materials-14-06519-f007:**
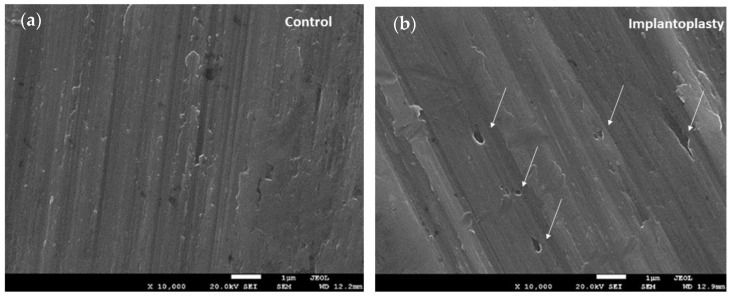
SEM images of the surfaces for (**a**) control and (**b**) implantoplasty samples.

**Table 1 materials-14-06519-t001:** Chemical composition of Hank’s solution.

Chemical Composition	NaCl	KCl	Na_2_HPO_4_	KH_2_PO_4_	CaCl_2_	MgSO_4_	NaHCO_3_	C_6_H_12_0_6_
Concentration (mM)	137	5.4	0.25	0.44	1.3	1.0	4.2	5.5

**Table 2 materials-14-06519-t002:** Results of the nanoindentation test performed on dental implants.

Load, (mN)	Mean Indentation Depth (SD), (mm)	Mean Hardness (SD), (GPa)	Mean Elastic Modulus (SD), (GPa)
1	89 (3)	2.89 (0.39)	70 (5)
2	135 (5)	2.28 (0.44)	66 (4)
3	168 (5)	2.56 (0.33)	65 (3)
4	197 (5)	2.53 (0.33)	65 (3)

Abbreviations: SD = Standard deviation.

**Table 3 materials-14-06519-t003:** Mechanical properties obtained by nanoindentation and residual stresses determined by X-ray diffraction. The results are expressed as mean and standard deviation.

Samples	Mean Hardness (SD), (GPa)	Mean Elastic Modulus (SD), (GPa)	Max Deformation (SD), (%)	Residual Stress (SD), (MPa)
Control disks	2.2 (1.2)	65 (5)	12.0 (4.2)	−27.5 (5.2)
Implantoplasty	4.8 (1.0)	80 (9)	4.3 (0.7)	−354.5 (35.2)
Ti cold-worked disks	4.7 (0.9)	78 (8)	4.0 (0.5)	−345 (3.2)

**Table 4 materials-14-06519-t004:** Electrochemical parameters obtained by the potentiodynamic curves for control and implantoplasty samples. The results are expressed as mean ± standard deviation.

Samples	Ecorr (SD), (mV)	Icorr (SD), (μA/cm^2^)	Polarization Resistance (SD), (Ω/cm^2^)	Corrosion Rate (SD), (mm/Year)
Control disks	−340 (32)	0.051 (0.007)	1.14 × 10^6^ (1.13 × 10^5^)	4.44 × 10^−4^ (6.69 × 10^−5^)
Implantoplasty	−368 (47)	0.055 (0.005)	1.07 × 10^6^ (1.77 × 10^5^)	4.77 × 10^−4^ (4.46 × 10^−5^)

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
