# Peer review of "Mechanical Properties and Corrosion Behavior of Ti6Al4V Particles Obtained by Implantoplasty: An In Vitro Study. Part II"

_materials, 2021, doi:10.3390/ma14216519_

Round 1

Reviewer 1 Report

- Add more current references (2020 – 2021).

-Improve the introduction with data about titanium alloys from the literature.

- You say on Table 3. Mechanical properties obtained by nanoindentation, and residual stresses determined by X-ray diffraction. How did you obtain the residual stresses determined by X-ray diffraction?

- There is a lot of research related to Ti6Al4V alloys, make a comparison between the results obtained by you and other previous research.

- Make the work uniform. All units of measurement please add them in square brackets: Table 2, 3 and 4.

- Improve the quality of Figure 6. Potentiodynamic curves for control and implantoplasty samples. The figure is unclear.

- Add more conclusions

- Generally the quality of the writing could be improved.

Author Response

Dear Reviewer,

Thanks for taking the time to review our manuscript and suggest to us to improve our work by providing a lot more detail. We have done so, and we are now submitting a manuscript that not only addresses the points the you specifically raised but also many others that we have considered in order to deliver what we think is a much improved version of our work. This version includes more paragraphs, English grammar revisions in all main sections, new references. Thanks a lot. We are looking forward to your comments.

Sincerely,

Francisco-Javier Gil Mur

  1. Add more current references (2020 – 2021).

In accordance with your suggestions, some references have been added within this scope of time.

  1. Improve the introduction with data about titanium alloys from the literature.

We have improved the introduction with data about titanium alloy, as suggested by the reviewer.  The following paragraph has been added in the Introduction section:

Commercially pure (c.p) titanium (Ti) dental implants are an excellent long-term treatment for patients by loss of teeth [1-2]. Titanium dental implants (grades I-IV) are highly reliable due to their excellent biocompatibility, mechanical characteristics, and good corrosion resistance among other properties [3-5].  However, the strongest grade of c.p. Ti has a strength of around 550MPa, then it has been designed different Ti alloys in order to increase the strength of the material. Among Ti alloys, Ti-6Al-4V is a (α- β)-type that has been used in a wide range of biomedical purposes. This alloy was firstly developed in the aerospace industry, but due to its strength, excellent corrosion resistance and biocompatibility Ti-6Al-4V is used for biomedical purposes. There are some concerns related to the long-term use of Ti-6Al-4V that have already been outlined in Part I. In summary, as Ti-6Al-4V contains toxic V, different alloys such as Ti–6Al–7Nb, Ti–5Al–2.5Fe or Ti-Zr have been proposed to replace it. However, Ti-6Al-4V remains the most widely used Ti alloy for dental implants.

  1. You say on Table 3. Mechanical properties obtained by nanoindentation, and residual stresses determined by X-ray diffraction. How did you obtain the residual stresses determined by X-ray diffraction?

A new paragraph has been introduced, in this text there is an short explanation about the method used to obtain the residual stresses. It’s the conventional and the most precise method for this determination

  1. There is a lot of research related to Ti6Al4V alloys, make a comparison between the results obtained by you and other previous research.

Following the suggestion performed by #reviewer 1 we have added a paragraph in the discussion section in order to compare our results with other previous publications.  

  1. Make the work uniform. All units of measurement please add them in square brackets: Table 2, 3 and 4.

Thank you for your assistance.  We have made the changes suggested by the reviewer.

  1. Improve the quality of Figure 6. Potentiodynamic curves for control and implantoplasty samples. The figure is unclear.

Originals figures have been introduced with more resolution.

  1. Add more conclusions

Thank you for your assistance. We have added more conclusions following the reviewer’s suggestion.

  1. Generally the quality of the writing could be improved.

Manuscript has been reviewed by an English editor. 

Author Response

document attached 

Reviewer 3 Report

The article “Mechanical properties and corrosion behaviour of titanium alloy particles obtained by implantoplasty: An in vitro study. Part II” is related to the study of the corrosion and mechanical properties of the particles of titanium alloy released during implantoplasty of dental implants. This is a continuation of the research of their properties presented in Part I. There are some comments.

  1. The grains of alpha phase (white phase) and beta phase (dark phase) are observed in the micrograph of the particles shown in Figure 2b. How the authors defined it? More detailed matallographic analysis is required.
  2. The article does not contain data on elemental analysis of the particles carried out with help of EDX method.
  3. In the caption to Figure 2, position b is missing. In addition, positions a and b are missing in Figures 1, 5, 6 and 7.
  4. What is the difference between the samples M2, M4 and M5 in the Figure 5 (left), as well as between the samples M3, M4 and M5 in the Figure 5 (right)?
  5. Two different groups of potentiodynamic curves are shown in Figure 6 (top), as well as in Figure 6 (bottom). What is the difference between these groups?
  6. The corrosion test performed using the bulk samples instead of particles is not correct because corrosion behavior depends not only on the phase composition of the material but on the active surface area of the samples. Particles with a large contact surface will react with electrolyte more actively.

Author Response

document attached

Round 2

Reviewer 1 Report

The manuscript under the title: “Mechanical properties and corrosion behaviour of titanium alloy particles obtained by implantoplasty: An in vitro study. Part II.” is relevant for the Materials. Paper discuss about corrosion behavior of titanium alloy particles released during implantoplasty of dental implants treated due to periimplantitis. A standardized protocol was used to obtain metal particles produced during polishing the surface of Ti-6Al-4V dental implants.The article based on original experimental research. The purpose of this investigation was to describe the mechanical characteristics and the corrosion behavior of metal particles originated from commonly used Ti-6Al-4V (grade 5) following an implantoplasty procedure. The organization of the article is appropriate. The abstract is sufficiently informative. Overall, the paper is well prepared. 

-English language and style are fine/minor spell check required

Author Response

Thank you for your remark. Authors have been reviewed the manuscript for English language spelling.

Reviewer 2 Report

1. Please correct the recognition in Figure 6.

Author Response

Thank you for your assistance. We have corrected the recognition in Figure 6.

Reviewer 3 Report

No comments

Author Response

Thank you very much for your help and consideration.